# DisCo-Layout: Disentangling and Coordinating Semantic and Physical Refinement in a Multi-Agent Framework for 3D Indoor Layout Synthesis

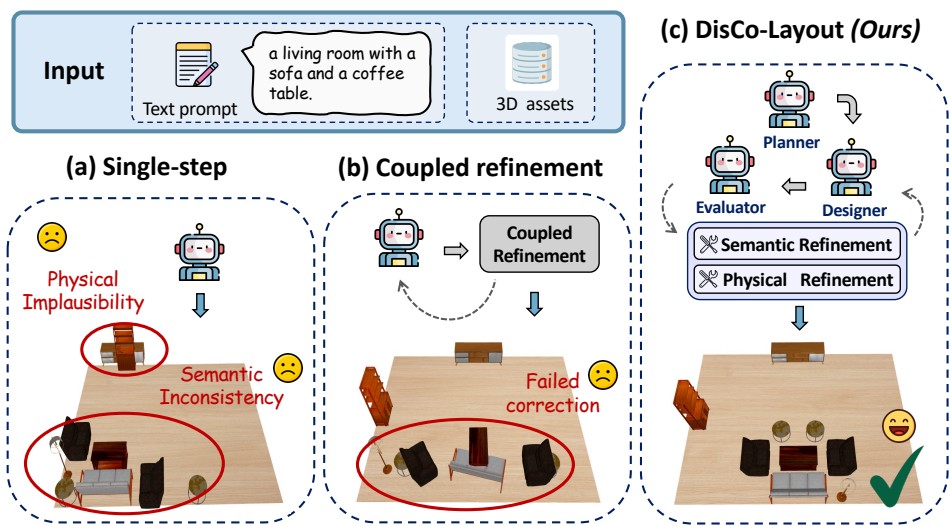

Figure 1: **Architecture comparison of LLM/VLM-based methods for 3D indoor scene synthesis. (a) Single-step methods** directly predict layouts without refinement, leading to inconsistent results in complex scenarios. **(b) Coupled refinement methods** integrate physical and semantic refinement in a coupled process, causing interference and limiting flexibility. **(c) The proposed DisCo-Layout** introduces a multi-agent framework that disentangles and coordinates semantic and physical refinement, enabling iterative and adaptive synthesis through collaboration between planner, designer, and evaluator agents.

## Abstract

3D indoor layout synthesis is crucial for creating virtual environments. Traditional methods struggle with generalization due to fixed datasets. While recent LLM and VLM-based approaches offer improved semantic richness, they often lack robust and flexible refinement, resulting in suboptimal layouts. We develop **DisCo-Layout**, a novel framework that disentangles and coordinates physical and semantic refinement. For independent refinement, our *Semantic Refinement Tool (SRT)* corrects abstract object relationships, while the *Physical Refinement Tool (PRT)* resolves concrete spatial issues via a grid-matching algorithm. For collaborative refinement, a multi-agent framework intelligently orchestrates these tools, featuring a *planner* for placement rules, a *designer* for initial layouts, and an *evaluator* for assessment. Experiments demonstrate DisCo-Layout's state-of-the-art performance, generating realistic, coherent, and generalizable 3D indoor layouts. Our code will be publicly available.

# 1 INTRODUCTION

3D indoor layout synthesis is driving advancements in various fields, including virtual reality (Patil et al., 2024), video games (Hu et al., 2024), and embodied AI (Krantz et al., 2020; Nasiriany et al., 2024; Kolve et al., 2017). This process involves creating a full 3D environment based on a natural language prompt and a library of 3D assets. The goal is to produce layouts that are not only physically plausible but also semantically aligned with the given instructions. The capacity to convert abstract, high-level commands into realistic, interactive 3D environments is becoming increasingly vital as downstream applications demand more automation and user-specific customization.

Traditional methods (Paschalidou et al., 2021; Tang et al., 2024; Yang et al., 2024a), which learn from predefined layout datasets (Fu et al., 2021), are consequently confined to in-domain 3D assets and fixed placement schemes. This limitation hinders their ability to generalize to unseen objects, novel layouts, or open-ended user instructions, reducing utility for real-world simulations and interactive environments.

The rise of large language models (LLMs) and vision-language models (VLMs) has unlocked exciting new avenues in 3D layout synthesis. By tapping into their understanding of spatial and object relationships, LLM-based approaches (Feng et al., 2023; Yang et al., 2024b; Sun et al., 2025a; Ling et al., 2025) create varied and semantically rich layouts directly from natural language prompts, even when working with open-domain 3D assets. For example, these models are able to deduce that chairs should face a table or bookshelves should align with walls.

However, despite this progress, several significant limitations persist: (i) *Lack of systematic laxyout refinement*: Some methods (Feng et al., 2023; Yang et al., 2024b) directly reason object placements using a single LLM or VLM without any refinement. As a result of these models' inherent randomness and tendency to "hallucinate," the generated layouts are inconsistent, physically implausible, or semantically nonsensical (Figure 1(a)). (ii) *Inflexible refinement strategies*: Other approaches attempt to refine layouts but employ intertwined mechanisms for both physical and semantic aspects (Figure 1(b)). For instance, LayoutVLM (Sun et al., 2025a) uses a unified differentiable optimization, which makes it challenging to balance competing objectives effectively. Similarly, TreeSearch-Gen (Deng et al., 2025) relies on VLMs to jointly assess physical and semantic criteria, often leading to interference between goals and causing suboptimal outcomes.

To tackle these limitations, we present **DisCo-Layout**, a novel framework that **Dis**entangles and then **Co**ordinates semantic and physical refinement through a VLM-based multi-agent system for 3D indoor **Layout** synthesis. This design enables the independent and collaborative refinement process, resulting in realistic and coherent 3D indoor layouts, as illustrated in Figure 1(c). Specifically, to disentangle these two refinement capabilities, DisCo-Layout first incorporates two specialized tools: the *Semantic Refinement Tool (SRT)*, which corrects high-level abstract object relationships (*e.g.*, chairs facing tables) via feedback-driven adjustment, and the *Physical Refinement Tool (PRT)*, which addresses low-level concrete spatial issues (*e.g.*, asset collisions) using a grid-matching algorithm. This separation ensures modularity and prevents conflicts between semantic and physical criteria. Furthermore, DisCo-Layout employs a multi-agent system to coordinate these tools with three specialized VLM-based agents: a *planner*, which derives high-level placement rules; a *designer*, which predicts initial 2D coordinates; and an *evaluator*, which assesses the layout for further refinement. This dedicated collaboration dynamically orchestrates the refinement process.

Extensive experiments demonstrate that DisCo-Layout achieves state-of-the-art performance, generating realistic, coherent, and physically plausible 3D indoor layouts. It also demonstrates robust generalization capabilities across diverse assets and natural language instructions, establishing a new baseline for 3D indoor layout synthesis.

Our main contributions are as follows:

- We present DisCo-Layout, a novel approach to 3D indoor layout synthesis that disentangles and coordinates physical and semantic refinement through specialized tools and a multi-agent system, producing realistic, coherent, and physically plausible layouts.

- We formulate two distinct refinement mechanisms: the Semantic Refinement Tool (SRT), designed to address abstract object relationships, and the Physical Refinement Tool (PRT), engineered for precise spatial adjustments.

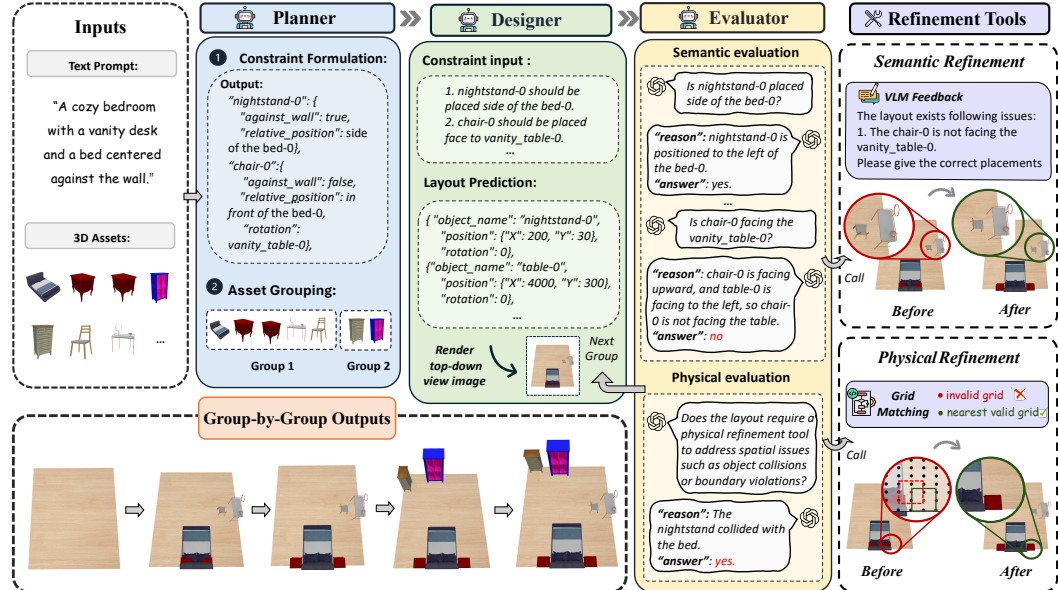

Figure 2: **Pipeline of DisCo-Layout.** Our method employs a multi-agent framework consisting of a planner, designer, and evaluator (Section 3.2). The planner determines placement rules based on asset properties, the designer predicts layout configurations for asset groups, and the evaluator assesses and refines the scene using a tool-use approach. Further, the refinement process is decoupled into the Semantic Refinement Tool (SRT) and Physical Refinement Tool (PRT), ensuring both contextual coherence and spatial consistency in the synthesized 3D indoor layouts (Section 3.3).

- We develop a VLM-based multi-agent framework comprising a planner for deriving high-level layout rules, a designer for generating initial layouts, and an evaluator for assessing layout quality and guiding refinement.

- Through extensive experiments, we demonstrate that DisCo-Layout achieves superior semantic accuracy and physical plausibility in 3D indoor layout synthesis compared to current state-of-the-art methods.

## 2 RELATED WORKS

### 2.1 3D INDOOR LAYOUT SYNTHESIS

3D indoor layout synthesis have evolved considerably. Early learning-based approaches (Paschalidou et al., 2021; Tang et al., 2024; Yang et al., 2024a) acquire robust layout representations from data and integrate differentiable physical constraints to generate floor plans. In contrast, other approaches exploit the spatial commonsense knowledge of large language models (LLMs) to synthesize realistic 3D layouts in a more direct and interpretable manner. For example, LayoutGPT (Feng et al., 2023) translates natural language descriptions into a 3D asset layout and then populates the layout by retrieving assets from 3D datasets (Fu et al., 2021; Deitke et al., 2023). Holodeck (Yang et al., 2024b) extends this by incorporating physical constraint optimization over the scene graph, enabling object placement that aligns more closely with human spatial intuition in open-vocabulary scene generation. LayoutVLM (Sun et al., 2025a) combines visually grounded VLMs with differentiable physical optimization strategies. TreeSearchGen (Deng et al., 2025) enhances the spatial reasoning capabilities of VLMs through hierarchical planning (Pun et al., 2025; Sun et al., 2025b). However, these methods still struggle to effectively disentangle semantic and physical refinement, often leading to suboptimal results. In our DisCo-Layout, we address this limitation by developing two specialized tools to separately handle semantic and physical refinement, enabling more coherent and realistic 3D indoor layouts.

## 2.2 MULTI-AGENT SYSTEM

Multi-agent systems aim to leverage distributed intelligence to solve complex tasks through agent collaboration (Luo et al., 2025). Traditional multi-agent systems are typically designed for lightweight agents with rule-based or function-call mechanisms (Rana & Stout, 2000; Deters, 2001). Recently, the advent of Large Language Models (LLMs) enables the creation of LLM-based agentic frameworks that can perform reasoning, planning, and dynamic collaboration at a much higher level of complexity (Wu et al., 2024; Li et al., 2023; Hong et al., 2023). LLM-powered multi-agent systems are applied to solve various real-world problems across different domains, such as scientific discovery (Ghafarollahi & Buehler, 2025; Chen et al., 2023), social simulation (Park et al., 2023), and visual content editing (Lin et al., 2025). In the domain of 3D indoor layout synthesis, previous works (Çelen et al., 2024; Wang et al., 2024; Hu et al., 2024) have also leverages multi-agent systems to tackle this challenging task, showcasing their initial potential in handling collaborative design processes. Following this paradigm, we design a multi-agent system to coordinate semantic and physical refinement, further incentivizing the collaborative capability of multiple agents in 3D indoor layout synthesis.

## 3 METHODOLOGY

The task of 3D indoor layout synthesis is formulated as generating a plausible layout for a set of objects within a constrained space. Given a natural language prompt $T$, a set of retrieved 3D assets $O = \{o_1, \ldots, o_n\}$, and a room defined by boundaries $B_{\text{room}} = \{b_1, b_2, b_3, b_4\}$, our goal is to produce a set of poses $P = \{p_1, \ldots, p_n\}$ where each pose $p_i = (x_i, y_i, \theta_i)$ specifies the position and rotation for the object $o_i$. The layout $P$ must satisfy two criteria: (1) *semantic coherence*, aligning with the functional and spatial relationships in $T$, and (2) *physical plausibility*, ensuring all objects are placed within $B_{\text{room}}$ without any collisions.

### 3.1 OVERVIEW

As shown in Figure 2, DisCo-Layout is built upon a multi-agent system comprising the planner, designer, evaluator and two specialized tools. The *planner* groups 3D assets and derives placement rules from their semantic and physical properties. Based on these constraints, the *designer* proposes initial coordinates and orientations to form a coarse layout. The *evaluator* then assesses the layout through visual question answering (VQA) over the rendered image from both semantic and physical aspects to determine whether the refinement is required. Furthermore, two dedicated refinement tools are leveraged to improve layout quality (Section 3.3). The *Semantic Refinement Tool (SRT)* ensures semantic consistency by correcting high-level relational violations, while the *Physical Refinement Tool (PRT)* enforces physical plausibility through precise geometric adjustments such as resolving collisions and out-of-bound placements using a grid-matching algorithm. These tools can operate either independently or jointly, allowing for fine-grained and flexible optimization. We first describe the three agents in Section 3.2, then in Section 3.3 the refinement tools are introduced.

### 3.2 MULTI-AGENT SYSTEM

DisCo-Layout utilizes a multi-agent system to coordinate semantic and physical refinement processes. This system consists of three specialized agents: a *Planner*, a *Designer*, and an *Evaluator*, which work collaboratively to handle different aspects of the layout generation process. By simulating how humans plan, design, and evaluate layouts in real-world scenarios, this framework ensures both modularity and flexibility in handling 3D indoor layout synthesis.

**Planner.** This agent is responsible for understanding the input instructions and generating high-level semantic placement rules to guide the layout synthesis process. Specifically, it performs two steps: (i) defining fine-grained constraints $C$ for all assets, and (ii) forming semantic asset groups and determining their placement order $G$, which is formalized as

$$(G, C) = f_{\text{Planner}}(T, O). \tag{1}$$

First, for every asset $o_i \in O$, it predicts a set of spatial constraints denoted as $c_i = \{c_{\text{wall}}, c_{\text{rel}}, c_{\text{rot}}\}$, which can be classified into two distinct categories. One category is *high-level semantic constraints*,

which govern the abstract relational logic of the scene. This category includes $c_{\text{rel}}$ defining the desired spatial relationship with other assets, and $c_{\text{rot}}$ specifying a target object to be faced (*e.g.*, a chair faces a table). Another category is *precise geometric alignment constraints*, represented by the boolean flag $c_{\text{wall}}$, indicating whether the object must adjoin a wall. Furthermore, we define six flexible relationship types for $c_{\text{rel}}$, consisting of *near*, *side of*, *in front of*, *aligned with*, *opposite*, and *around*. These constraints guide the designer in clustering related assets into coherent zones (*e.g.*, a dining area) and supply the evaluator with ground truth information for semantic verification and refinement.

Subsequently, based on these constraints, the planner partitions the asset set $O$ into a set of disjoint ordered groups $G = \{G_1, G_2, ..., G_K\}$, where the order of the groups reflects their relative priority in the layout synthesis process. Assets that are physically co-located, semantically associated, or symmetrical (*e.g.*, a pair of night-stands) are typically assigned to the same group. This group-by-group strategy ensures that primary, room-defining objects are positioned before ancillary furniture, enabling more targeted and effective refinement at each iteration.

**Designer.** For each semantic group $G_k$ produced by the planner, the designer is responsible for generating an initial layout for each asset $o_i \in G_k$ and their associated constraints $c_i$, which is expressed as

$$\tilde{P}_k = f_{\text{Designer}}\left(T, \{(o_i, c_i)\}_{o_i \in G_k}, I_{k-1}\right), \tag{2}$$

where $\tilde{P}_k = \{\tilde{p}_i \mid o_i \in G_k\}$ is the set of output poses, and $I_{k-1}$ is the top-down rendered image of the existing layouts prior to placing group $G_k$. Leveraging the VLM's ability to process high-level instructions, we find that simultaneously placing all assets from the same group enhances group semantic coherence, as it excels at generating layouts with correct spatial relationships, such as placing chairs next to tables. However, due to the inherent limitations of VLMs, the predicted layout may result in suboptimal placements that fail to meet semantic and physical constraints. These issues are delegated to the refinement process for resolution.

**Evaluator.** The evaluator assess the predicted layout for both semantic coherence and physical plausibility. The assessment is formalized as

$$(s_{\text{sem}}, s_{\text{phy}}) = f_{\text{Evaluator}}(P'_k, I'_k, \{c_i\}_{o_i \in G_k}), \tag{3}$$

where $P'_k = P_{k-1} \cup \tilde{P}_k$ represents the updated global layout after integrating the proposed layout $\tilde{P}_k$ for $G_k$ with existing layout $P_{k-1}$, and $I'_k$ is the corresponding rendered image. The output $(s_{\text{sem}}, s_{\text{phy}})$ comprises two booleans indicating whether the current layout requires semantic or physical refinement. To perform this evaluation, the module employs a unified VQA-based approach for both semantic and physical aspects, querying a vision-language model (VLM) with targeted questions. For semantic coherence, the assessment focuses only on high-level relational constraints, *i.e.*, $c_{\text{rel}}$ and $c_{\text{rot}}$, which are translated into corresponding yes-or-no questions. For instance, the constraint *"the chair is placed near the table"* is transformed into the question *"Is the chair placed near the table?"*. If any question receives a "no" response, the Semantic Refinement Tool would be employed to address the issues. For physical plausibility, the evaluator identifies whether the current layout requires adjustments at the coordinate level. If the VLM detects issues such as object collisions or out-of-bounds placements, the Physical Refinement Tool is invoked to make the necessary adjustments. This evaluation approach reduces the potential for erroneous or uncontrolled judgments by narrowing the VLM's focus, and it enforces semantic consistency by ensuring the evaluator and planner adhere to the same placement standards.

## 3.3 Refinement Tools

To address issues in the initial layout predictions, we employ two refinement tools designed to resolve layout issues and thus enhance layout quality. These tools are triggered when the evaluator detects semantic or physical problems. Specifically, we integrate two disentangled refinement tools, the *Semantic Refinement Tool (SRT)*, which focuses on correcting high-level relationships, and the *Physical Refinement Tool (PRT)*, which resolves physical issues such as object collisions. Figure 3 illustrates how SRT and PRT enhance layout plausibility.

**Semantic Refinement Tool.** This tool resolves semantic issues in the layout, focusing on incorrect object relationships (*i.e.*, $c_{\text{rel}}$) or improper orientations (*i.e.*, $c_{\text{rot}}$). Its goal is to make precise

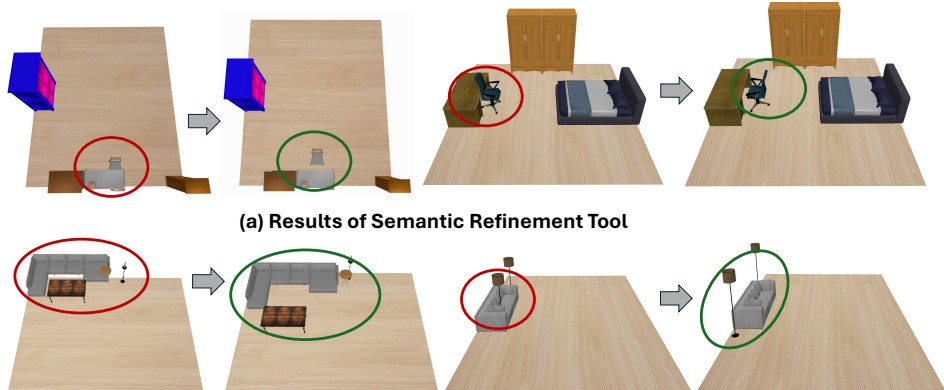

(a) Results of Semantic Refinement Tool

qi tmm

(b) Results of Physical Refinement Tool

Figure 3: **Visualization of the correction effects of our SRT and PRT.** The top row (a) showcases the result of SRT, which focuses on semantic coherence. The bottom row (b) demonstrates the result of PRT, which enforces physical plausibility.

adjustments without disrupting the overall layout. This refinement process is formalized as

$$P_j'' = \begin{cases} P_j' & \text{if } s_{\text{sem}} = \text{False}, \\ f_{\text{SRT}}(P_j', C_{\text{sem}}^*) & \text{if } s_{\text{sem}} = \text{True}, \end{cases} \quad (4)$$

where $C_{\text{sem}}^*$ is the set of failed semantic constraints and $P_j''$ is the updated layout. The core of our approach is to convert each failed constraint in $C_{\text{sem}}^*$ into an explicit text. For instance, if the evaluator identifies that a chair was not facing a table, the failed constraint is transformed into a direct, natural language feedback *"The chair is not facing the table"*. The feedback is forwarded to a VLM, which then proposes a minimal position or rotation adjustment that resolves the issues without disturbing the overall layout.

**Physical Refinement Tool.** This tool corrects coordinate-level violations within the semantically validated layout. Its output is the definitive layout $P_j$ for the current iteration, which is determined as:

$$P_j = \begin{cases} P_j'' & \text{if } s_{\text{phy}} = \text{False}, \\ f_{\text{PRT}}(P_j'') & \text{if } s_{\text{phy}} = \text{True}. \end{cases} \quad (5)$$

An object is considered invalid if coordinate-based checks reveal that it meets any of the following conditions: (i) its bounding box overlaps with another object's bounding box, (ii) it is not entirely contained within the room boundaries, or (iii) it violates $c_{wall}$. To resolve these issues, we introduce an innovative grid matching algorithm designed to reposition the object to a feasible grid point. The algorithm begins by discretizing the floor plan into a uniform grid set $\mathcal{M} = \{m_1, m_2, m_3, \ldots, m_n\}$. For each invalid object, it determines the nearest valid grid point $m_i$ for repositioning. A valid grid point must simultaneously satisfy three key criteria: (i) *Collision-Free*: the repositioned object must have zero Intersection over Union (IoU) with other objects, ensuring no collisions occur.; (ii) *Within Boundaries*: The repositioned object must be fully enclosed within the room's boundaries, without exceeding its limits; and (iii) *Aligned with Walls*: When the property $c_{wall}$ is activated, the repositioned object's back face must align precisely with the wall nearest to its center coordinates. Once the optimal grid point is identified, the object's coordinates are updated accordingly. Requiring a candidate grid point to satisfy all these criteria simultaneously, the algorithm guarantees that a single corrective placement resolves all physical violations without introducing new ones. This discrete, grid-based approach offers a fast, stable, and precise solution to physical constraints, ensuring minimal positional changes that preserve existing semantic relationships. More details are provided in the Appendix.

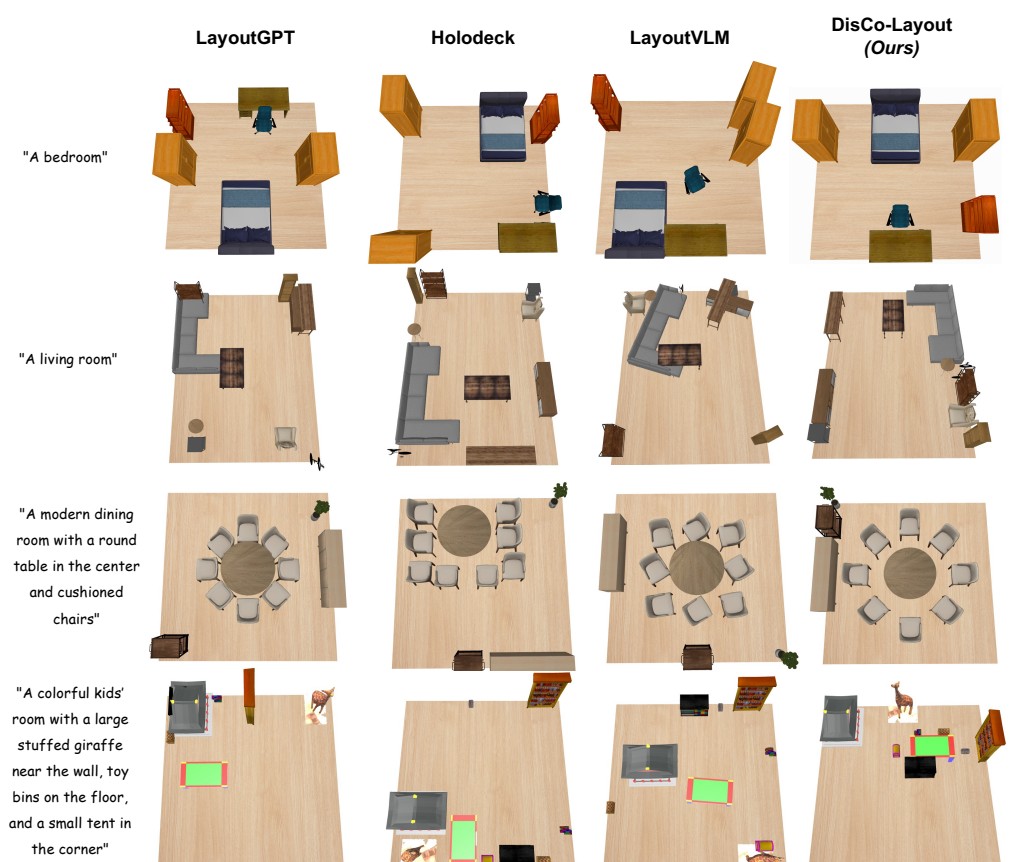

Figure 4: **Qualitative comparison.** Using the same prompts and assets as inputs, Disco-Layout demonstrates its ability to generate semantically and physically plausible layouts, while accurately reflecting the spatial intent of the given prompts.

Table 1: **Quantitative comparison.** Our DisCo-Layout achieves *zero* physical violations across all categories while maintaining top semantic accuracy, achieving overall state-of-the-art performance.

| Method | Bathroom | | | | Bedroom | | | | Dining Room | | | | Kitchen | | | | Living Room | | | |
| | Physical | | Semantic | | Physical | | Semantic | | Physical | | Semantic | | Physical | | Semantic | | Physical | | Semantic | |
| | Col.↓ | OOB↓ | Pos.↑ | Rot.↑ | Col.↓ | OOB↓ | Pos.↑ | Rot.↑ | Col.↓ | OOB↓ | Pos.↑ | Rot.↑ | Col.↓ | OOB↓ | Pos.↑ | Rot.↑ | Col.↓ | OOB↓ | Pos.↑ | Rot.↑ |
|---|---|---|---|---|---|---|---|---|---|---|---|---|---|---|---|---|---|---|---|---|
| LayoutGPT | 5.13 | 18.86 | **72.6** | 47 | 4.84 | 17.38 | 66 | **70.2** | 10.40 | 7.62 | 79 | 75.2 | 5.14 | 33.43 | 50.2 | 57 | 4.36 | 11.27 | 65.8 | 68.6 |
| Holodeck | **0.00** | **0.00** | 62 | 55 | **0.00** | 2.22 | **74.2** | 68 | **0.00** | **0.00** | 79.4 | 70.4 | **0.00** | **0.00** | 57.6 | 45.8 | **0.00** | **0.00** | **74.8** | 61.6 |
| LayoutVLM | 17.89 | 16.22 | 54 | 43.83 | 10.00 | 9.52 | 60.8 | 66 | 5.47 | 3.20 | 71 | 61 | 5.08 | 16.29 | 50.6 | **61.2** | 9.87 | 5.64 | 73.2 | 67.2 |
| Ours | **0.00** | **0.00** | 68 | **66** | **0.00** | **0.00** | 74 | 69 | **0.00** | **0.00** | **87.8** | **88.5** | **0.00** | **0.00** | 57.2 | 48 | **0.00** | **0.00** | 69.6 | **74.2** |

| Method | Buffet Restaurant | | | | Classroom | | | | Children Room | | | | Home Gym | | | | Average | | | |
| | Physical | | Semantic | | Physical | | Semantic | | Physical | | Semantic | | Physical | | Semantic | | Physical | | Semantic | |
| | Col.↓ | OOB↓ | Pos.↑ | Rot.↑ | Col.↓ | OOB↓ | Pos.↑ | Rot.↑ | Col.↓ | OOB↓ | Pos.↑ | Rot.↑ | Col.↓ | OOB↓ | Pos.↑ | Rot.↑ | Col.↓ | OOB↓ | Pos.↑ | Rot.↑ |
|---|---|---|---|---|---|---|---|---|---|---|---|---|---|---|---|---|---|---|---|---|
| LayoutGPT | 2.34 | 8.46 | **64.2** | 65.2 | 1.37 | 20.52 | **67.8** | 60.6 | 7.57 | 18.16 | **74.6** | 63.6 | 4.76 | 26.41 | 60.8 | **63** | 5.10 | 18.01 | 66.78 | 63.38 |
| Holodeck | **0.00** | **0.00** | 56.2 | 47 | **0.00** | **0.00** | 59 | 55.2 | **0.00** | **0.00** | 58.8 | 64.6 | **0.00** | **0.00** | 61.2 | 56.2 | **0.00** | 0.25 | 64.8 | 58.2 |
| LayoutVLM | 6.50 | 3.33 | 62.2 | 52 | 3.44 | **0.00** | 64.6 | 59.8 | 14.65 | 8.57 | 64 | 55.4 | 8.95 | 14.05 | **69.4** | 62.2 | 9.09 | 8.54 | 63.31 | 58.74 |
| Ours | **0.00** | **0.00** | 64 | **69.8** | **0.00** | **0.00** | 64.8 | **66** | **0.00** | **0.00** | 67.6 | 65 | **0.00** | **0.00** | 58 | 55.4 | **0.00** | **0.00** | 67.89 | 66.88 |

# 4 EXPERIMENTS

## 4.1 SETUPS

**Datasets.** For comprehensive evaluation, we curated a test set of 45 indoor scenes, evenly spanning 9 categories. The categories are partitioned into five common (bathroom, bedroom, dining room, kitchen, and living room) and four uncommon (buffet-restaurant, classroom, children's room, and home gym) scenes, so that the benchmark concurrently probes routine and open-domain scenarios. For each scene, we employed GPT-4o to synthesize a natural language prompt ranging in complexity from succinct labels (*e.g.*, a living room) to intricate, multi-clause specifications (*e.g.*, a modern

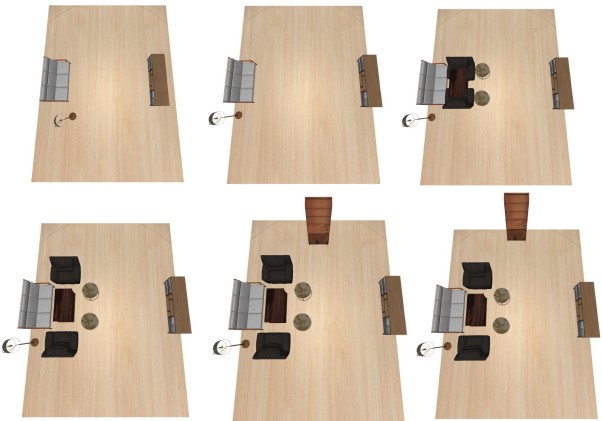

Figure 5: **Group-by-group visualization of our synthesized layout.** Disco-layout progressively places three groups of assets for a living room, while immediately refining semantic and physical errors at each stage.

kitchen with a central island surrounded by bar stools). This variance enables a controlled study of sensitivity to prompt specificity. Following the Holodeck protocol (Yang et al., 2024b), we retrieved candidate assets from Objaverse (Deitke et al., 2023) and manually screened them for semantic fidelity and geometric quality.

**Evaluation Metrics.** We evaluate generated layouts along two complementary axes: physical plausibility and semantic coherence. Physical plausibility is quantified with two geometric indicators (i) Collision Rate, the proportion of object pairs with intersecting bounding volumes, and (ii) Out-of-Bounds Rate, the share of an object's surface area that lies outside the prescribed room boundaries. Semantic coherence follows the LayoutVLM protocol(Sun et al., 2025a), employing GPT-4o to score: (i) Positional Coherency, which verifies that objects occupy functionally sensible locations (*e.g.*, a chair on the floor rather than on a bed), and (ii) Rotational Coherency, which checks that orientations afford expected usage (*e.g.*, chairs facing a table, a television directed toward the sofa).

**Compared Methods.** We evaluate DisCo-Layout against two categories of LLM-driven, text-conditioned 3D indoor scene synthesis baselines. The first category comprises methods lacking an explicit refinement process, including LayoutGPT and Holodeck. LayoutGPT prompts an LLM with in-context examples and directly predicts numerical coordinates for object positions and rotations. Holodeck conducts a floor-based search for object placements, where it first prunes locations that cause collisions or out-of-bounds errors and then weights the remaining valid candidates based on a predefined relationship graph. The second category features methods with a unified refinement phase, represented by LayoutVLM. LayoutVLM adopts a two-stage process, first using a VLM to predict initial coordinates and then formulating differentiable objective functions to jointly optimize for both physical and semantic goals.

## 4.2 Experimental Results

**Quantitative Results.** As presented in Table 1, our method achieves superior performance across both semantic and physical metrics. Baselines lacking a refinement stage, such as LayoutGPT and Holodeck, struggle to excel in both aspects simultaneously. While LayoutGPT can generate strong semantic associations by leveraging the commonsense knowledge of VLMs to reason object placements, it lacks a mechanism for fine-grained coordinate adjustment, resulting in poor physical plausibility. Conversely, Holodeck's object-by-object search strategy effectively avoids collisions and boundary violations, but its reliance on predefined rules and sequential placement limits its capacity to represent complex, globally coherent semantic relationships. Although LayoutVLM incorporates refinement, its unified optimization approach forces a trade-off between competing objectives, where improving one aspect can degrade the other, thus preventing it from achieving optimal results in both physical and semantic aspects. In contrast, DisCo-Layout's agent-based framework intelligently coordinates two disentangled tools, selectively applying targeted refinements to resolve specific flaws without mutual interference, thereby surpassing all baselines.

**Qualitative Results.** The qualitative comparison is illustrated in Figure 4, which contains diverse prompts of varying complexity, ranging from a simple description such as "A living room" to a detailed one with specific layout constraints like "A modern dining room with a round table in the center and cushioned chairs". The results show that our method generates layouts that are both physically plausible and semantically coherent; the PRT effectively maintains physical integrity, while the SRT organizes related objects into logical functional zones (*e.g.*, a bar cart can adjacent to the dining table). Figure 5 visualizes the step-by-step process of our method, highlighting how interleaved refinement enables the construction of complex, coherent, and physically plausible scenes. Although LayoutGPT can determine reasonable general locations, it suffers from severe collisions and boundary violations. Holodeck's sequential, object-by-object placement can lead to globally suboptimal arrangements (*e.g.*, a table against a wall, making chair placement illogical) and its rigid, predefined constraints prevent it from adhering to specific layout instructions in the prompt (*e.g.*, placing a small tent in a corner). LayoutVLM, struggling to balance its dual objectives, still produces layouts with significant object collisions and implausible object poses. Therefore, our method uniquely combines the VLM's powerful semantic reasoning for initial generation with a disentangled refinement process that minimally adjusts coordinates to ensure physical realism without corrupting the semantic integrity of the scene.

## 4.3 ABLATION STUDIES

To validate the effectiveness of our core design within the DisCo-Layout framework, we conduct a series of ablation studies, systematically dismantling our model to isolate and analyze the contribution of each core component. The variants are as follows: (i) w/o Semantic Refinement Tool (SRT), where only the PRT is available to correct physical errors; (ii) w/o Physical Refinement Tool (PRT), where only the SRT is available to correct semantic errors; (iii) w/o both, which removes the entire refinement stage (Evaluator, SRT, and PRT), taking the initial layout generated by the Designer as the final output. and (v) Evaluator with Open-Ended VQA, where we modify the Evaluator's diagnostic process from a structured VQA format to an open-ended question (*e.g.*, Could you identify any semantically unreasonable floor objects in this scene?).

Table 2 shows the results of each component. Removing either the SRT or PRT leads to a significant degradation in semantic and physical scores. Furthermore, replacing the structured VQA with open-ended questions results in a decline in semantic coherence, which suggests that the VLM's open-ended responses can be less focused and actionable, underscoring the importance of our carefully designed, targeted prompts for effective error diagnosis. Collectively, these findings validate our core hypothesis: the disentangled refinement architecture, intelligently coordinated by a specialized evaluator, is crucial for achieving high-quality, coherent 3D scene generation.

Table 2: **Ablation Study.** The full framework is compared with variants that ablate individual refinement components (*i.e.*, SRT and PRT), the entire refinement process, or are combined with open-ended VQA.

| Method | Physical | | Semantic | |
|---|---|---|---|---|
| | Col.↓ | OOB↓ | Pos.↑ | Rot.↑ |
| **w/o SRT** | 0.73 | 1.08 | 67.27 | 64.67 |
| **w/o PRT** | 12.21 | 10.21 | 66.22 | 63.13 |
| **w/o SRT & PRT** | 10.76 | 12.22 | 66.97 | 64.16 |
| **w/ OpenVQA** | 1.24 | 0.99 | 67.6 | 64.4 |
| **DisCo-Layout** | **0.00** | **0.00** | **76.16** | **76.72** |

## 5 CONCLUSION

This paper investigates the task of 3D indoor layout synthesis, which involves generating fully arranged 3D environments from a text prompt and a set of 3D assets, while ensuring both physical plausibility and semantic coherence. We formulate DisCo-Layout, an approach that disentangles and coordinates physical and semantic refinement for 3D indoor layout synthesis with two dedicated tools and a multi-agent framework. Experimental results demonstrate that DisCo-Layout achieves state-of-the-art performance, producing realistic and coherent 3D indoor layouts in both physical and semantic aspects. In the future, we aim to further enhance the framework's generalization to more complex scenarios and to improve the training of methods for embodied AI.

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

# A  APPENDIX

## A.1  DETAILS OF DISCO-LAYOUT

In this section, we provide the details of our method including the prompts adopted by three specialized agents and Semantic Refinement Tool (SRT). We also explain the procedure of the grid matching algorithm.

PROMPT OF THE PLANNER

```
You are a professional interior designer who can create reasonable
    layouts based on the room type and furniture list provided.

You need to complete the task according to the following steps:

1. For each floor object, consider its global position (whether it's
    against a wall) along with its relative position and
    orientation to other objects, then organize the information in
    json format which contains the following keys:

- against\_wall: Whether the object is placed against a wall.
    Positioning objects against walls improves space efficiency, and
    most objects in a room (e.g., nightstands, desks, cabinets,
    sofas, floorlamps) should be placed this way.
- relative\_position: The positional relationship between this
    object and others. Grouping functionally related objects
    together helps form cohesive functional zones, making the layout
    more compact and organized.
- relative\_object: All objects referenced in relative\_position.
    Please select words from [near, side of, in front of, aligned
    with, opposite, around] to describe positional relationships.
- rotation: The object that this one should face (e.g., chairs
    should face tables).  Only give the target object name.

2. Based on the object relationships you provided, group these
    objects together. A semantic asset group is a collection of
    assets that are logically related to each other. The grouping of
    the assets are based on functional, semantic, geometric, and
    functional relationships between assets.
Usually assets that are close to each other in space can be grouped
    together. For example, a bed, a nightstand can be grouped
    together.
However, it is also possible to group assets that are not physically
    close to each other but are semantically related. For example,
    a sofa, a tv console in front of the sofa can be grouped
    together even though the tv and the tv console is a few meters
    away from the sofa. They can be grouped together because they
    are semantically related -- the tv is in front of the sofa.
Besides, multiple objects of the same category are usually grouped
    together, which enhances their symmetry when placed. For example
    , coffee table-0, armchair-0, armchair-1, side table-0, side
    table-1 can be grouped together.

Then, you will order the groups based on the sequence in which they
    should be placed in the scene. You should consider the
    significance of each group and the logical flow of the scene
    layout. For example, larger or more prominent assets may be
    placed first to establish the scenes focal points.

Please organize your answer in the following json format:

```json
```

```
\{"constraints": \{"sofa-0": \{
      "against\_wall": true, "relative\_position": null,
      "rotation": null \}
      ,...\}

    "groups":\{"group1": ["sofa-0", "tv stand-0", "floot lamp
      -0"],\}...\}
```

Now you need to design \{room\_type\}, here is the object list: \{
    object\_names\}.

Note:

1. Do not consider the impact of entrances, doors, or windows.

2. I prefer placing furniture such as bathtub, floorlamp, plant
    stand against walls to save space.

3. Strictly output in JSON format without adding any extra content
    before or after.

4. If the object is against the wall, its rotation needs to be null.

PROMPT OF THE DESIGNER

```
You are an interior design expert working in a 2D space using an X,
    Y coordinate system. (0, 0) represents the bottom-left corner of
     the room. All objects must be placed in the positive quadrant,
    meaning their coordinates should be positive integers in
    centimeters. By default, objects face the +Y axis.

You are designing {room_type} with dimensions {room_size}. Below is
    a JSON file containing a list of existing objects in the room:
{arranged_objects}

with the following details:
- object_name: The name of the object. Follow the name strictly.
- size: Considering the size of the object helps prevent collisions
    when predicting coordinates.
- position: The coordinates of the object (center of the object's
    bounding box) in the form of a dictionary, e.g., {{"X": 120, "Y
    ": 200}}.
- rotation: The object's rotation angle in the clockwise direction
    when viewed along the z-axis toward the origin, e.g., 90. The
    default rotation is 0, which aligns with the +Y axis.
Please select rotation from 0, 90, 180, or 270 degrees, where a 0-
    degree object faces upward, a 90-degree one faces right, a 180-
    degree one faces downward, and a 270-degree one faces left.

You will also receive the corresponding rendered top-down layout
    view featuring brown floors surrounded by white walls.
You need to place objects {current_group} according to the following
     constraints:
{constraint_strings}

Remember:
- Place objects in the area to create a meaningful layout. Objects
    layout should be align with common human perceptions of the room
```

```
        's function, usage habits, spatial efficiency, or aesthetic
        principles in terms of rationality and logic.
    - When arranging closely placed objects (e.g., chairs around a table
        , beds with nightstands on both sides), their dimensions must be
         accounted for to avoid collisions.
    - For objects placed against the wall, consider half of their length
         or width to determine their coordinates, while ensuring they
        are rotated to face the center of the room.
    - I prefer to place large floor objects such as sofas, beds, and
        cabinets against the wall, and their rotation angle should be
        set to face the center of the room.
    - Based on the room's layout, only provide the object_name, size,
        position, and rotation of the new added objects in JSON format.
    - All provided items must be placed within the room.
    - Only generate JSON code; do not include any other content.
    - Strictly follow the provided object_name without any modifications
        .
```

PROMPT OF THE EVALUATOR

```
    You are an interior designer. Given a rendered image of a room, you
        need to observe the image and answer my questions.

    Scene Coordinate Specifications:

    - You only need to consider the 2D space of the floor.
    - Use the X,Y coordinate system. The X-axis represents the
        horizontal direction, the Y-axis represents the vertical
        direction, and (0, 0) denotes the origin at the bottom-left
        corner.

    You will receive a top-down view rendered image of a room featuring
        brown floors surrounded by white walls, along with information
        about the floor objects present (object\_name, size, position,
        rotation).

    - object\_name: The name of the object. Follow the name strictly.
    - size: The size of the object. Follow the size strictly.
    - position: The coordinates of the object (center of the object's
        bounding box) in the form of a dictionary, e.g., \{"X": 120, "Y
        ": 200\}.
    - rotation: The object's rotation angle in the clockwise direction
        when viewed along the z-axis toward the origin, e.g., 90. The
        default rotation is 0, which aligns with the +Y axis. (This
        means that a 0-degree object faces upward, a 90-degree one faces
         right, a 180-degree one faces downward, and a 270-degree one
        faces left.)

    Now, this is a top-down view image of \{room\_type\} with dimensions
        \{room\_size\} showing the objects \{objects\_in\_image\}.
    You need to answer the following questions based on the image layout
        and their positions and rotations:
    \{questions\}

    Please organize your answer in the following format:

    ```json
    [
        \{
```

```
            "question": "Is chair-0 placed in front of table-0, align
                with table-0?",
            "objects":['chair-0','table-0'],
            "reason": give your reason,
            "answer": only answer yes or no
        \{,
    ...
    ]
    ```

    Follow these instructions carefully:

    - When determining whether the rotation is correct, please refer to
        the rules: ** a 0-degree object faces upward, a 90-degree one
        faces right, a 180-degree one faces downward, and a 270-degree
        one faces left. **
    - Do not add any additional text at the beginning or end.
    - Do not change the object names.
```

```
    You are an interior design expert. Given a top-down view rendering
        of an indoor layout, where the position and rotation of
        furniture are directly predicted by a VLM (Vision-Language Model
        ), you need to determine whether the furniture layout requires
        micro-level physical adjustments to achieve a more realistic
        arrangement. If furniture is not flush against walls (if needed)
        , goes out of bounds, or has collisions, it usually requires
        fine-tuning of position and rotation.

    Below is an image of a {room_type}. Please determine whether it
        requires a physical refinement tool to optimize the layout.
    Output only True or False without additional text.
```

PROMPT OF THE SEMANTIC REFINEMENT TOOL

```
    You are an interior design expert working in a 2D space using an X,
        Y coordinate system. (0, 0) represents the bottom-left corner of
         the room. All objects must be placed in the positive quadrant,
        meaning their coordinates should be positive integers in
        centimeters. By default, objects face the +Y axis.

    You are designing \{room\_type\} with dimensions \{room\_size\}.
        Below is a JSON file containing a list of existing objects in
        the room:
    \{objects\_in\_image\}
    with the following details:
    - object\_name: The name of the object. Follow the name strictly.
    - size: The size of the object. Follow the size strictly.
    - position: The coordinates of the object (center of the object's
        bounding box) in the form of a dictionary, e.g., \{"X": 120, "Y
        ": 200\}.
    - rotation: The object's rotation angle in the clockwise direction
        when viewed along the z-axis toward the origin, e.g., 90. The
        default rotation is 0, which aligns with the +Y axis.
    You have identified the following object placement issues: \{result
        \}. Please output the corrected JSON based on the corrected
        positions.
```

```
Remember:
Only provide the JSON of the object that needs to be modified.
The JSON keys should include object\_name, size, position, and
    rotation.
you only generate JSON code, nothing else. It's very important.
    Respond in markdown.
```

GRID MATCHING ALGORITHM

In this subsection, we present the grid matching algorithm used in the Physical Refinement Tool (PRT), with the detailed procedure shown in Algorithm 1.

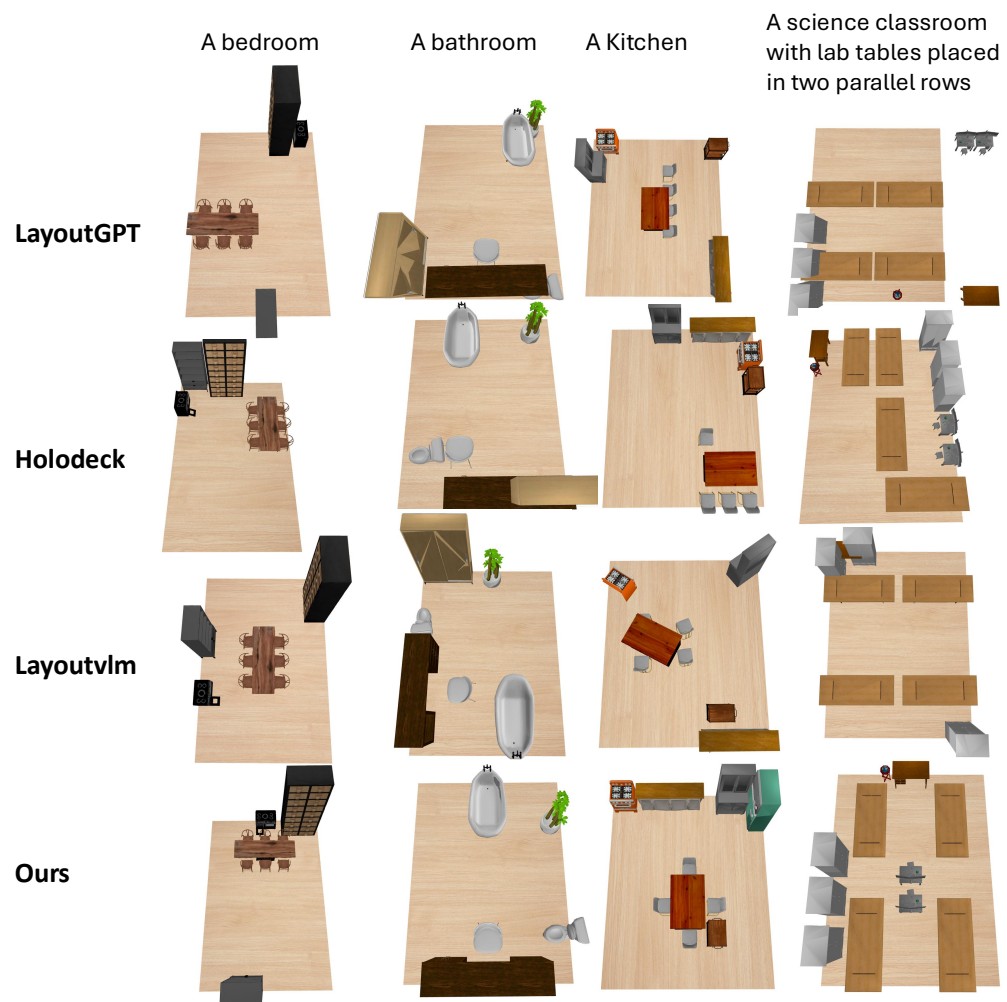

Figure 6: **More qualitative results comparing DisCo-Layout with baselines.** These extended comparisons highlight our model's consistent ability to produce more physically plausible, and semantically coherent layouts.

## A.2 MORE QUALITATIVE RESULTS

This section provides additional visual results to demonstrate the robustness and versatility of our method. In Figure 6, we showcases diverse scenes generated from more text prompts, comparing our

---

**Algorithm 1** Grid Matching Algorithm

---

1: **Input:** semantically validated layout $P_j'' = \{p_1, \ldots, p_n\}$; room boundaries $B_{\text{room}} = [b_1, b_2, b_3, b_4]$; grid set $\mathcal{M} = \{m_1, \ldots, m_k\}$
2: **Output:** final layout $P_j$ of the current iteration
3: /* **Step 1 – Align with walls** */
4: **for all** $p_i'' \in P_j''$ **do**
5:     **if** is_against_wall($p_i''$, $B_{\text{room}}$) **then**
6:         $b^\star \leftarrow$ find_nearest_wall($p_i''$.position, $B_{\text{room}}$) // $b^\star \in [b_1, b_2, b_3, b_4]$
7:         $p_i''$.rotation $\leftarrow$ wall2rotation($b^\star$)
8:         back_center $\leftarrow$ get_back_center($p_i''$)
9:         grid_list $\leftarrow$ sort_grid_by_distance(back_center, $\mathcal{M}_{b^\star}$)
10:         **if** grid_list is empty **then**
11:             delete $p_i''$ from $P_j''$
12:         **else**
13:             nearest_grid $\leftarrow$ grid_list[0]
14:             $p_i''$.position $\leftarrow$ pull_to_wall(nearest_grid, $p_i''$, back_center)
15:         **end if**
16:     **end if**
17: **end for**
18: /* **Step 2 – Out-of-Bounds Correction** */
19: OOB $\leftarrow$ find_oob_objects($P_j''$, $B_{\text{room}}$)
20: **for all** $p_i'' \in$ OOB **do**
21:     grid_list $\leftarrow$ sort_grid_by_distance($p_i''$.position, $\mathcal{M}$)
22:     grid_list $\leftarrow$ filter_grid_if_OOB(grid_list, $p_i''$, $B_{\text{room}}$)
23:     **if** grid_list is empty **then**
24:         delete $p_i''$ from $P_j''$
25:     **else**
26:         nearest_grid $\leftarrow$ grid_list[0]
27:         $p_i''$.position $\leftarrow$ nearest_grid
28:     **end if**
29: **end for**
30: /* **Step 3 – Collision Resolution** */
31: collision_pairs $\leftarrow$ find_collision_pairs($P_j''$)
32: **for all** $\langle p_u'', p_v'' \rangle \in$ collision_pairs **do**
33:     smaller_object $\leftarrow$ compare_size($p_u''$, $p_v''$)
34:     grid_list $\leftarrow$ sort_grid_by_distance(smaller_object.position, $\mathcal{M}$)
35:     grid_list $\leftarrow$ filter_grid_if_collision(grid_list, smaller_object, $P_j''$)
36:     **if** grid_list is empty **then**
37:         delete smaller_object from $P_j''$
38:     **else**
39:         nearest_grid $\leftarrow$ grid_list[0]
40:         smaller_object.position $\leftarrow$ nearest_grid
41:     **end if**
42: **end for**
43: **return** $P_j$ // *After in-place updates on $P_j''$, denote the result as $P_j$.*

---

outputs with those from key baselines. These extended comparisons highlight our model's consistent ability to produce more physically plausible and semantically coherent layouts.

## A.3 THE USE OF LARGE LANGUAGE MODELS (LLMS)

In this work, Large Language Models (LLMs) were used as a general-purpose assist tool for grammar correction and language polishing. Specifically, the LLM was employed to assist in refining the writing and ensuring clarity in language. It helped with grammatical adjustments, rephrasing sentences for better readability.

