# OpenReview forum: "DisCo-Layout: Disentangling and Coordinating Semantic and Physical Refinement in a Multi-Agent Framework for 3D Indoor Layout Synthesis"
_ICLR.cc/2026/Conference — ICLR 2026 Conference Withdrawn Submission_

### Official Review · Reviewer_dkPL · 2025-10-31

**Soundness:** 2
**Presentation:** 3
**Contribution:** 1
**Rating:** 2
**Confidence:** 4

**Summary:**

This paper presents a method for synthesising furniture layouts for rectangular rooms. It uses pretrained VLMs to assemble a multi-agent pipeline – a planner decides a set of furniture and constraints on placement; a designer refines this to exact layouts; an evaluator checks whether all constraints are satisfied; and a refinement stage improves placements where objects are not in an ergonomically-sensible arrangement. The evaluator and refinement tools are used to provide feedback to the designer, improving its layout iteratively. The method is tested on 45 prompts for nine room types. Quantitative results exceed existing LLM/VLM-based furniture layout methods.

**Strengths:**

The overall pipeline, i.e. the combination of agents, their respective tasks, and how they communicate, is novel. The idea of using VLMs for layout generation is a sensible one, and having separate critics is also logical. The separation of semantic from physical concerns (i.e. whether objects are placed ergonomically to form a functional room, versus whether there are interpenetrations, floaters, etc.) is a good idea.

Experiments are conducted on diverse prompts for nine room types, including several unusual ones for which the benefit of pretrained VLMs is clearer (since large training datasets would be hard to obtain). For evaluation, both physical and semantic correctness are measured; the latter is scored by another VLM. Quantitative results against three fairly recent baselines (LayoutGPT, Holodeck and LayoutVLM) show that the proposed method generally out-performs them across room types.

Prompts vary in levels of detail, some simplify specifying a room type, others giving details of the furniture. In general adherence to the prompts is fairly good in the qualitative examples presented in the manuscript, and better than the selected LLM/VLM-based baselines in these cases.

There is an ablation study measuring the benefit of certain components, showing that each of them yields a noticeable contribution either to semantic or physical quality.

The paper is clear, well-structured, and pleasant to read.

**Weaknesses:**

This is a complex pipeline that yields underwhelming results. In particular, the qualitative results in Figure 4 are greatly lacking realism – half rooms left empty, sparse layouts, and objects placed somewhat randomly around the edge of the room. Results are considerably worse than Merrell's work from 2012, and other more recent methods that are learnt from datasets (ATISS, etc.). Claims of "realism" and "coherence" are manifestly too strong. While I appreciate the proposed approach is intended to be training-free, these datasets and methods nonetheless exist and should be seen as defining state-of-the-art.

There are no related works prior to 2021 referenced, despite various methods existing dating back to at least 2011 (e.g. Interactive furniture layout using interior design guidelines, Merrell, SIGGRAPH 2011; Make it home: automatic optimization of furniture arrangement, Yu, TOG 2011; Automatic Generation of Constrained Furniture Layouts, Henderson CoRR 2017; Human-centric indoor scene synthesis using stochastic grammar, Qi CVPR 2018), among others.

The actual technical contribution is fairly small. Multi-agent systems for refining layouts etc. are now fairly well established, and the specific extensions here are small (without interesting technical novelties) and very domain-specific (hence not of great interest to the broader community). The overall 'recipe' for the pipeline also does not seem to yield many interesting or transferable insights for general readers in the ICLR community.

**Questions:**

The problematic issues that require addressing are mentioned under "Weaknesses" above. In particular, for me to favor acceptance, the paper would have to demonstrate considerably more impressive results, particularly given the complexity of the pipeline.

It would also be good to describe genuine technical contributions in the introduction (i.e. fundamentally novel techniques, insights, etc.), rather than just "we built a pipeline" and "we evaluated it".

---

### Official Review · Reviewer_QBuM · 2025-11-01

**Soundness:** 2
**Presentation:** 3
**Contribution:** 2
**Rating:** 2
**Confidence:** 4

**Summary:**

This paper presents DisCo-Layout, a framework for room layout generation. DisCo-Layout uses several VLMs: a planner, which derives high-level placement rules; a designer, which predicts initial 2D coordinates; and an evaluator, which assesses the layout for further
refinement. It also develops the Semantic Refinement Tool (SRT) and Physical Refinement Tool (PRT) to correct local errors made by the designer. The authors evaluate the method on 45 indoor scenes and claim some improvements over baselines.

**Strengths:**

1. The paper is well-written and easy to follow. All components are clearly described.
2. Splitting the agents into planner and designer makes sense. Grouping furniture into groups and optimizing their locations together also seem to be a good idea.

**Weaknesses:**

- Problem with the evaluation of the proposed method:
1) room layout has a high degree of freedom and is very hard to judge automatically. In this paper, no human evaluation provided, and the qualitative results are very limited.
2) Even out of the limited qualitative results, results are not promising. For example, Fig4, row4: the bottom half of the room is basically empty, which is not realistic.
3) In general, for all the test cases presented in the paper, I find the prompts being not specific enough, and the constraints being too easy to fulfill given the large size of the room and very few furniture to put in. These make the proposed method seem unnecessarily complicated, as the constraints are likely to be fulfilled by some heuristic-based methods (such as the ones used in Infinigen [1]), without even involving VLMs.
Given the presented results, I'm not convinced that the proposed method is better than baselines such as Holodeck. I suggest that the authors to provide human studies, and provide more visualizations on rooms which much more complicated instructions.

- Why is the evaluator necessary? From my understanding, the evaluator checks if the orientations of the placed furniture are correct, and if they are collision-free and lie within the room boundary. I think all these can be easily checked by hand-designed rules, why do you need VLM invlolved here?

- Semantic refinement and Physical Refinement: the refinement could solve collisions when the rooms are very empty, but it becomes way harder to resolve when the placement of furniture becomes compact. It could be impossible for the model to adjust the local placement to satisfy all constraints. I.e., some global re-adjustment would be necessary at some moment. This global adjustment is common when humans design their room layout. Does your method have any stopping criteria when the refinement fails, and does it have any fall-back plans?

- Overall, the contribution and the novelty of the paper are limited. The papers chain several off-the-shelf VLM agents together, which is not novel. The core components such as Physical Refinement is basically placing objects on regular grids to avoid intersections, and seems to be a liitle bit too trivial to be a solid contribution.


- (minor) typo: L074 laxyout -> layout

[1] Infinigen Indoors: Photorealistic Indoor Scenes using Procedural Generation

**Questions:**

N/A

---

### Official Review · Reviewer_b2DC · 2025-11-01

**Soundness:** 3
**Presentation:** 3
**Contribution:** 2
**Rating:** 4
**Confidence:** 5

**Summary:**

DisCo-Layout proposes a multi-agent framework for indoor layout synthesis: a planner derives grouping and placement order from text and available assets; a designer produces initial object poses conditioned on a top-down view; and an evaluator uses structured VQA to check semantic and physical compliance. If criteria are not met, two decoupled refinement tools are invoked—SRT (semantic refinement) to minimally correct relations/orientations, and PRT (physical refinement) that uses grid matching to remove collisions/out-of-bounds issues and align objects to walls when needed. This generate → evaluate → targeted (semantic/physical) refinement loop improves physical feasibility while preserving semantic consistency. Experiments across various room types and baselines show zero physical violations with equal or better semantic scores, and ablations confirm the necessity of both refiners and the evaluator.

**Strengths:**

1.	The method follows a transparent sequence—planning → group-wise layout generation → evaluation → semantic refinement → physical refinement—with well-defined module boundaries, which makes it easy to reproduce and extend.
2.	Semantic errors are handled only by the SRT, and physical errors only by the PRT, so the VLM focuses on a single objective per step, reducing interference in model judgments.
3.	Room-defining, large objects are placed first, followed by ancillary and decorative items. This ordering allows each group to be refined in a targeted way, avoiding unnecessary global rearrangements.
4.	The PRT discretizes the floor into valid grid positions and “snaps” violating objects to feasible cells, which reliably removes collisions, out-of-bounds cases, and wall-attachment violations, and is more controllable than open-ended generative adjustments.

**Weaknesses:**

1.	The pipeline assumes the availability of structured 3D assets (with geometry/bounds), which limits applicability when such assets are absent or incomplete.
2.	Heavy reliance on VLM/LLM components. Planner, designer, and evaluator all depend on the capability and stability of the underlying model; the paper does not thoroughly analyze robustness to weaker or shifted models.
3.	One-way semantic→physical pipeline. The system first fixes semantics and then applies physical refinement; the physical stage selects the nearest valid grid but does not re-validate semantics afterward. In cases where multiple physically valid placements exist but only a subset preserves the intended semantic relations (e.g., chair-side arrangements around a table), the current design cannot guarantee the semantically preferable choice.
4.	Limited experimental scale. Experiments are mainly on 9 categories and 45 scenes, which shows promise but is still small for strong claims about generalization to richer room layouts.
5.	No vertical/small-object reasoning. Evaluation is essentially 2D/top-down; vertical relations and placement of small items (on tables, in shelves, stacked)

**Questions:**

1. How long does it take to generate a room, and how many tokens are consumed? End-to-end generation time and token count might be acceptable for a single pass, but the step-by-step setting needs more detailed reporting.

2. How strong is instruction following? If a user specifies spatial relations directly in the input, what results do you obtain?

3. In principle, agentic approach should perform better in irregular rooms. Can you evaluate some cases?

---

### Official Review · Reviewer_A3DQ · 2025-11-02

**Soundness:** 2
**Presentation:** 2
**Contribution:** 2
**Rating:** 2
**Confidence:** 4

**Summary:**

The paper proposes a multi-agent framework for 3D scene layout synthesis. It involves collaborations between three agents, namely planner, designer and evaluator. The overall pipeline is fairly straightforward, and the proposed method seems to work effectively for the examples shown. However, the tasks performed by individual agents are fairly straightforward, and often considered in earlier works (apart perhaps not using VLM), so the technical novelty of the method is limited. Relying on a multi-stage process could lead to the system being less robust, as failure in one step could lead to failure of the whole system (there are some mechanisms for correction such as the evaluator, but even this agent can make mistakes).

**Strengths:**

The overall pipeline is reasonable and appears to produce plausible results, by exploiting the capabilities of VLMs.

**Weaknesses:**

The design is fairly straightforward, and the individual agents are mimicking steps in traditional or deep learning pipelines. The collaboration between agents is fairly limited, and there are no theoretical guarantee that mistakes can be effectively corrected. The paper thus has limited technical novelty.

Having a pipeline with multiple (fragile) steps could mean the overall system is not robust.  For example, although the file refinement can help address some issues, there are no guarantees that a plausible solution can be found as changes may have to be applied from the very beginning of the coarse layout).

The layout is represented as an image for the VLM. However, it cannot effectively handle cases where objects overlap from the top-down view, some of which are valid and others may not be so (e.g. objects can be placed on a table).

The individual agents are nearly hard-coded for 3D indoor layout synthesis, and the proposed pipeline has limited potential inspiration for broader applications.

There are also quite a few typos in the paper, e.g. on Page 3, 3D indoor layout synthesis have => ... has.

**Questions:**

How does the method handle failures of individual agents?

What is the success rate in practice?

---

### Note · Authors · 2025-11-27

I have read and agree with the venue's withdrawal policy on behalf of myself and my co-authors.